# Development of the Biofidelic Instrumented Neck Surrogate (BINS) with Tunable Stiffness and Embedded Kinematic Sensors for Application in Static Tests and Low-Energy Impacts

**DOI:** 10.3390/s25164925

**Published:** 2025-08-09

**Authors:** Giuseppe Zullo, Elisa Baldoin, Leonardo Marin, Andrey Koptyug, Nicola Petrone

**Affiliations:** 1Department of Industrial Engineering, University of Padua, 35131 Padova, Italy; elisa.baldoin.1@studenti.unipd.it; 2Department of Mechanical Engineering, University of Utah, Salt Lake City, UT 84112, USA; leonardo.marin@utah.edu; 3Department of Quality and Mechanical Engineering, Mid Sweden University, 831 25 Östersund, Sweden; andrey.koptyug@miun.se

**Keywords:** neck surrogate, dummy neck, neck injury, neck brace, anthropomorphic test device, injury criteria, impact analysis

## Abstract

**Highlights:**

Neck surrogates could be helpful tools to assess injury risk in simulated impacts but still can be perfectionated to capture the complexity and diversity of neck anatomy and behavior in different situations. The Biofidelic Instrumented Neck Surrogate (BINS) aims to offer a tool for the analysis of low-energy impacts and also to introduce a novel set of sensors to investigate neck movements without external equipment. This works explores the mechanical design of this surrogate as well as the working principle of the embedded sensors, providing comparison with literature data and proof of the sensor concept in static and dynamic tests. The BINS is still far from catching the whole complexity of the human neck behavior, but offers a valuable tool in low-energy scenarios, enriched by the possibility of getting quite accurate head/trunk relative kinematics by embedded sensors.

**What are the main findings?**
The BINS surrogate has similar flexural behavior to passive human subject.BINS embedded angular sensors return neck flexural and compressive measurements with similar performances with respect to motion capture cameras.

**What is the implication of the main finding?**
BINS neck could be used as a viable tool to obtain realistic head kinematics in low-energy impacts.The developed angular sensors could be used effectively to quantify BINS movements and could be adapted to future neck surrogates structurally designed to withstand high-energy impacts or crashes.

**Abstract:**

Road accidents could result in severe or fatal neck injuries. A few surrogate necks are available to develop and test neck protectors as countermeasures, but each has its own limitations. The objective of this study was to develop a surrogate neck compatible with the Hybrid III dummy, focused on tunable flexural stiffness and integrated angular sensors for kinematic feedback during impact tests. The neck features six 3D-printed surrogate vertebral bodies interconnected by rubber surrogate discs, providing a baseline flexibility to the surrogate fundamental spinal units. An adjustable inner cable and elastic elements hooked on the sides of vertebral elements allow to increase the flexural stiffness of the surrogate and to simulate the asymmetric behavior of the human neck. Neck flexural angles and axial compression are measured using a novel system made of wires, pulleys, and rotary potentiometers embedded in the neck base. A motion capture system and a load cell were used to determine the bending and torsional stiffness of the neck and to calibrate the sensors. Results showed that the neck flexural stiffness can be tuned between 3.29 and 5.76 Nm/rad. Torsional stiffness was 1.01 Nm/rad and compression stiffness can be tuned from 39 to 193 N/mm. Sensor flexural angles were compared with motion capture angles, showing an RMSE error of 1.35° during static testing and of 3° during dynamic testing. The developed neck could be a viable tool for investigating neck braces from a kinematic and kinetic perspective due to its inbuilt sensing ability and its tunable stiffness.

## 1. Introduction

Protecting people in sports, work, and traffic has always been a priority for health organizations and traffic regulators. Traffic-related injuries account for most deaths due to head trauma [1]. Cervical spine injuries are a concern as well because of their potentially severe consequences. Two-wheeled vehicle users are at high risk [2,3], but sports can also pose people at risk [4]. Generally, strategies to mitigate traffic injuries involve the enforcement of protective equipment and the definition of safety requirements for both vehicles and gear.

Crucial to this point is the availability of testing devices for assessing the effectiveness of safety measures. This has led to the development of Anthropomorphic Test Devices (ATDs), such as the Hybrid III dummy [5]. ATDs are intended to model a human with surrogate body segments that behave as closely as possible to a reference human subject. Therefore, a full-body ATD could be used as a dummy occupant for testing vehicles during experimental crash simulations, or some of its sub-systems such as head/neck complex could be used to test specific protective gear (e.g., neck braces and helmets), offering human-like response without the need for post-mortem human subjects.

On this matter, the availability of a biofidelic neck surrogate is of great importance, because it can provide an accurate prediction of spinal injuries and is also responsible for correct head kinematics. Nevertheless, modeling this part of the human body is challenging, not only because of its anatomical complexity, but also because muscular activation drastically changes its dynamic and kinematic behavior. The measurement of human neck is the starting point to design such surrogates, like data collected from volunteer subjects or cadaveric specimens (e.g., vertebrae and ligaments) [6,7,8,9]. The former provided stiffness data of conscious neck and the latter failure loads for the cervical biological structures.

Subsequently, several neck surrogates were produced using these data, each with different purposes and technical specifications. These surrogates can be grouped into three main categories: (i) beam-like surrogates, simulating the neck with a multidirectional flexible beam; (ii) functionally equivalent surrogates, with a design not necessarily similar to the human neck anatomy, but with the same kinematics and stiffness; (iii) modular surrogates, with multiple interconnected elements mimicking the spinal units. Among the first type, the Hybrid III ATD has a flexible neck with asymmetric flexion and extension and has been validated for frontal car crash impacts. Some neck surrogates were later inspired by the Hybrid III. The BioRID I was developed to simulate rear and low-velocity car crashes [10]. Moreover, Walsh et al. proposed an unbiased alternative to Hybrid III, maintaining the same geometry and construction characteristics, but without asymmetric behavior [11]. Another variation of the Hybrid III design was reported in ISO 13232, in which a neck with an adjustable static flexion angle was used in a motorcycle crash test dummy [12].

In addition to these examples, different designs of the Hybrid III neck have recently been reported. MacGillivray et al. produced a beam-like neck surrogate to be mounted on the Hybrid III head form, compared it with the Hybrid III neck during low-speed impacts, and focused on the effect of this new neck on the resulting head kinematics [13]. Farmer et al. published a biofidelic surrogate neck with different flexion/extension and lateral bending behaviors [14]. Their neck features ball joints and several flexible bands to provide an adequate range of motion and stiffness; therefore, it stands in the functionally equivalent neck subgroup. Finally, an example of a modular neck is the surrogate patented by Rueda-Arreguín, Ceccarelli, and Torres-Sanmiguel, which uses springs and elastic elements to control the angular displacement between rigid vertebrae [15,16]. However, little is known about the structural resistance of the neck and its applicability to full-scale impact tests. This neck also features Inertial Measurement Units (IMU) sensors that provide a measurement of the angles after some signal processing. The use of IMU for angle measurements was also investigated with human subjects and compared to optoelectronic Motion Capture, obtaining good, but not perfect, agreement in slow movements [17]. While the abovementioned surrogate focused on flexural behavior, Nelson and Cripton addressed the high stiffness of the Hybrid III neck in head-first axial impacts, proposing a modular surrogate with a more biofidelity response with respect to cadaver tests [18].

Based on existing literature, some problems still need to be addressed, since none of the existing surrogates is able to provide a biofidelic response in multiple anatomical planes and for different degrees of muscular activation and neither allows for the collection of kinematic data during an impact. Although neck angles aren’t strictly needed for neck injury assessment, since main criteria are based on kinetic measurements [19,20], future criteria could also rely on failure related to exceeding kinematic thresholds [8]. Moreover, kinematics of the neck could also be meaningful to analyze the head movement during impacts and crash experimental simulations.

Based on these evidences, the present work aimed to develop a biofidelic dummy neck having a bending, torsional and axial response similar to human cervical spine. The developed surrogate presents a modular design and tuning elements that allow separating stiffness modification for each spinal unit and plane, allowing to behave differently in flexion, extension and lateral bending as in the human cervical spine. The surrogate also integrates angular sensors to provide neck angles in the sagittal and lateral planes. Finally, the surrogate is compatible with the Hybrid III dummy, allowing full-scale tests with this ATD, and to measure both the occipital and C7 neck loads using neck multiaxial load cells.

## 2. Materials and Methods

The object of this study was the development of a biofidelic neck surrogate, the Biofidelic Instrumented Neck Surrogate (BINS), characterized by a modular design, tunable stiffness, and integrated sensors. An overall graphical description of the BINS is given in Figure 1.

During the conceptualization of the BINS, a first requirement was the compatibility with the Hybrid III head and trunk, which set the outer dimensions of the surrogate as well as the design for the top and bottom connections. Further requirements were the modularity (to allow for length scaling) and the tunability (to mimic different neck stiffnesses and muscle contribution): this required to separate the surrogate into six vertebral components. This allowed each unit to be tuned separately from the others, thus replicating the anatomy and physiology of the human cervical spine [8]. As final requirements, the neck shall be designed to include kinematic and load sensors to capture the relative motion between dummy trunk and head, as well as occipital loads and moments.

### 2.1. Mechanical Design

The developed surrogate comprises six independent Fundamental Spinal Units (FSUs). Each FSU is inspired by its biological counterpart and is composed of two rigid plastic components connected by a rubber cylinder ideally representing the vertebrae and the intervertebral disc, respectively. The surrogate vertebrae are 3D printed in Polyamide 12 using selective laser sintering and have a footprint circumscribed in an 85 mm circle to be compatible with the Hybrid III head and trunk. Thanks to the high resolution and the additive process of selective laser sintering, it was possible to obtain inside the vertebral component a hollow cup (50 mm inner diameter, 6 mm depth) presenting a multi-layer orthogonal grid to bond with a rubber surrogate vertebral disc. Two vertebral components are joined together using a 50 mm diameter cylinder of PlatsilGel 10 silicone rubber (Polytek, Easton, PA, USA), mimicking the biological disc. Using a set of two molds, interposed between an upper and a lower vertebral component, the rubber is poured into the 50 mm diameter cavity and left to cure, ultimately connecting the two vertebral components and building an FSU. Indeed, the 3D printed vertebrae provide a structural support for the rubber and for the other elements composing the BINS. On the other hand, the rubber is responsible for achieving the proper viscous-elastic response of the surrogate in bending, torsion, and compression. Therefore, the material properties of the surrogate disc were targeted to obtain a neck bending response similar to that in cadaveric experiments [8]. More in detail, the rubber of the disc was selected after testing rubbers of different hardness and therefore elastic properties in preliminary experiments with single FSUs. Best results were achieved using A10 Shore rubber, like the product PlatsilGel 10 adopted for the full prototype. After preparation of the six FSU, these are connected one on top of the other by four M4 bolts and allow to build up the full neck surrogate together with the base and top pieces. About these two components, they are manufactured using the same technology for the vertebral bodies. The top piece replicates the connection between the Hybrid III neck and head form. The bottom piece connects the neck to the Hybrid III trunk and hosts the sensors and will be discussed more in detail in the next section.

The stiffness of the surrogate can be also modified independently in four directions (flexion, extension, and left/right lateral bending) owing to the four hooks placed on the frontal, backward, and lateral sides of the outer circumference, which allow the positioning of multiple elastic elements such as rubber O-rings. Indeed, in the present study the neck stiffness was modified by using nitrile rubber circular O-Rings sized 20.22 mm in diameter and with a circular section of 3.53 mm diameter. Moreover, a central hole of 5.5 mm allows the running of an inner cable (IC), which by its pre-tensioning controls the neck stiffness and range of motion, as in the Hybrid III surrogate. The IC is built using a 1.6 mm diameter steel cable brazed to an M5 threaded rod. The IC is fixed on the top surface and can be tensioned from the lower side by tightening the M5 nut to the threaded rod of the screw.

### 2.2. Sensors

Angular sensors for flexion/extension and lateral bending were embedded in the neck. These sensors measure the inclination of the upper surface of the neck in two planes with respect to its lower base. To obtain such measurements, each FSU features four conduits parallel from its axis of symmetry and equally distributed at a 14.5 mm distance from its center. Each conduit is coated with low-friction PTFE sheaths with an inner diameter of 1.5 mm (PTFE Tube Shop, Breda, The Netherlands). Conduits for each FSU align creating four continuous sheaths running from the top to the bottom of the neck. Inside these sheaths, four wires made of coated braided fishing line (Carp Fishing Line Coated Hook Link, Dongborui Outdoors Co., Ltd., Shenzhen, China) were fixed to the top of the neck and passed through the surrogate. The chosen wire has an outer diameter of 0.4 mm and a rated strength of 111 N. Upon reaching to the bottom piece of the neck, each wire is connected to a pulley embedded in the sensor case. Each pulley is spring-loaded by a torsional spring and spins together with a rotary potentiometer coupled to the same 4 mm diameter steel shaft. The shaft is connected to the sensor housing by a ball bearing minimizing friction. The detailed sensor assembly is shown in Figure 2.

To describe the working principle of the neck the fishing wires can be supposed to be inextensible. Indeed, their stiffness appeared qualitatively greater than that of the torsional spring and the two are connected in series, making the latter taking most of the deformation. Therefore, we approximated a change in length of the conduit to induce the sole rotation of the torsional spring (and pulley), neglecting any change of length of the wire itself. In any case, the calibration procedure will compensate for any deviation from this theoretical assumption since it will implicitly consider the variation in length of the wires.

The working principle of the sensor exploits the geometrical changes of the neck and in particular of the above-described conduits. During neck bending, opposite wires have longer and shorter paths available, and because the wires are inextensible, opposite pulleys are pulled by the wire and spring, respectively. This modifies the voltage reading from the potentiometer in proportion to the angular displacement with opposite signs between opposite potentiometers. Neck compression can be determined using the same set of sensors; in this case, each wire shortens equally, and compression can be linked to the sum of the voltage readings from the potentiometers. A graphical representation of sensors working principle is in Figure 3.

Moreover, the top surface of the neck allowed positioning of an M3564F six-axis load cell (Sunrise Instruments, Nanning, China) to measure the loads at the occipital level. Both the sensor base and load cell are shown in Figure 1 with the full neck assembly.

### 2.3. Testing

A setup similar to that of [8] was implemented to test the flexural behavior of the neck and the accuracy of the angular sensors. Briefly, the base of the surrogate was fixed while the occipital side of the neck was rotated by a pure bending moment. The moment is applied by means of two opposite and equal forces applied at the two ends of a 1 m aluminum beam, as shown in Figure 4a. The forces were generated using calibrated weights and a pulley system. The angle was measured using a Motion Capture system made of five Bonita infrared cameras (Vicon, Oxford, UK) based on the angular deflection of a triad of markers positioned over the top beam. The triad is composed by three markers defining the vertexes of an isosceles triangle lying on a plane parallel to the neck upper ending, with the basis aligned to the beam axis and centered on its origin. The moment was read using a K68D load cell (ME Systeme, Hennigsdorf, Germany) positioned at the bottom of the neck and acquired with a SoMat eDAQ lite (HBM, Darmstadt, Germany) synchronously with the four potentiometers. The moment to the neck was applied in progressive discrete steps and then removed backwards.

Bending tests on the BINS prototype were made once for each orientation (flexion/extension and lateral bending) and configuration (number of IC nut turns (i.e., pre-compression), and position and number of stiffening elements (O-Rings, OR)), for a total of 32 tests. Finally, a validation test in which the neck was freely moved by hand in a random manner was performed to assess the sensor calibration.

The same setup was then used to measure the torsional response of the BINS. In this configuration, the pulley system was moved to re-direct the forces to induce a torque and a rotation about the neck axis of symmetry.

In addition to the bending and torsional tests, an axial compression test was performed using an MTS242 (MTS, Minneapolis, MN, USA) servo hydraulic cylinder to compress the top of the neck while fixing the bottom end. The test was used to measure the axial stiffness of the neck during compression and calibrate the compression response of the angular sensors. The test setup is illustrated in Figure 4b.

Finally, an impact test was set up by mounting the BINS on a Hybrid III dummy and impacting the head of the dummy in the back with a steel pendulum set to impact energy of 20 J, as shown in Figure 4c. The selected impact energy was sufficient to provoke a large angular deflection (i.e., such that the helmet contacts the dummy trunk). The dummy was wearing an SM8 motorcycle helmet (Alpinestars, Asolo, Italy). The impact was captured using the motion-capture system and the SoMat eDAQ used in the calibration, both set to a sample rate of 250 Hz (the maximum available for the camera system). This test validated the possibility of using the BINS and its sensors in a full-scale low-energy impact test.

To achieve synchronization between the SoMat and the Vicon camera system, an external trigger closed a circuit which sent a digital signal to the SoMat and lit an infrared LED in the field of view of the camera. The digital signal started the SoMat acquisition and the motion capture footage was trimmed from the first frame in which the LED is visible.

### 2.4. Analysis

#### 2.4.1. Bending and Torsion Test

The data from the motion capture, load cell, and potentiometer voltages were imported into MATLAB R2024b. The three reflective markers on the top of the beam were used to build a reference system with the X-axis along the aluminum beam (sagittal), the Y-axis normal to the top of the neck (vertical), and the Z-axis as the cross-product of X and Y (lateral). This reference now identified by its 3 × 3 orientation matrix R was used as an input to rotm2eul function to calculate the Cardan ZXY angles of the upper neck vertebrae: X-angle was used for lateral bending; Y-angle for axial rotation; Z-angle for flexion/extension. Using the proper angle and moment data, the conventional stiffness was calculated as the slope of the linearized moment-angle response (least squares method) for flexion, extension, lateral bending, and torsion. The bending stiffness of the neck was tested for statistical significance with the built-in anovan function using the numbers of IC turns, Front O-Rings, Back O-Rings, and Lateral O-Ring pairs as grouping factors.

In the bending test, the raw signals from the potentiometers were combined by subtracting the opposite transducers. In this manner, the signal of the desired angle is amplified, and the cross-axis sensitivity of the two other sensors is reduced. The voltage output of the potentiometers (*V*) was then calculated as a function of the two angles (*θ*) in sagittal and frontal plane, and a calibration matrix for the neck was obtained. A least-squares linear fitting was used to obtain *K* from the equation below, and subsequently solve the overdetermined linear system made of the samples collected during the bending test.(1)$$\left(\theta_{flex/ext}\;\theta_{lat}\;\right)=\left[k_{11}\;k_{12}\;k_{21}\;k_{22}\;\right]\left(V_{front}-V_{back}\;V_{right}-V_{left}\;\right)$$

#### 2.4.2. Axial Test

In the axial test, both force and displacement data were returned by the hydraulic cylinder. The same least-square procedure was used to determine stiffness, but in this case two linear zones were individuated and two separate slopes were calculated.

To measure the axial compression (*d*) using the sensors, the sensor outputs (*V*) was combined using the following equation, after the calibration coefficient (*k_c_*) was determined via least-squares linear fitting similar to the bending test.(2)$$d=k_{c}\sum_{i=1}^{4}V_{i}$$

#### 2.4.3. Impact Test

Using a similar procedure to the bending test, the two triads of markers on the dummy head and trunk were used to build two reference systems: R_H_ and R_T_. Then, by matrix pre-multiplication by the transpose of themselves before the impact, these reference systems were aligned to the same XYZ directions such that both rotate about the same Z axis when the pendulum hits the dummy. Finally, the matrix product R_N_ = R_T_ ^T^∗R_H_ returned the rotation matrix between trunk and head, whose Cardan angle representation are the relative neck angles. Thanks to this procedure, any relative movement between head and helmet and/or rigid motion of the dummy trunk are removed. Therefore, the angles obtained with the motion capture system can be compared with the angles measured with the potentiometer system, obtained by processing the voltage data with the Equation (1).

## 3. Results

### 3.1. Characterization Tests

Results of bending tests for some configurations are reported in Table 1 together with literature data and showed statistically significant differences in bending stiffness due to number of IC turns (F = 23.07, *p* < 0.001), number of Front O-Rings (F = 17.3, *p* < 0.001), number of Back O-Rings (F = 52.55, *p* < 0.001), and number of Lateral O-Ring pairs (F = 8.5, *p* = 0.0017). The full list of configurations tested is available in Appendix A. Moreover, the bending response of the different tested configurations of the neck, other surrogates, and human reference data is shown in Figure 5. The figure reports data of the BINS with 0 and 6 turns of the IC (IC0, IC6), and of the BINS with 0 turns and O-Rings (IC0+OR, 1 Lateral pair and 2 Back O-Rings).

Torsional behavior of the neck is consistently stiffer than the human (cadaver) neck, but way lower than Hybrid III neck, with a torsional stiffness of 1.01 Nm/rad.

About the axial test, the compression response of the BINS can be approximated with a bilinear model. Indeed, the neck stiffness in the initial compression is of 39 N/mm, but after about 7 mm of compression, the BINS exhibits a stiffening behavior due to a degree bulging of the rubber disc exposed portion, increasing the stiffness to 193 N/mm. Therefore, adjusting the number of IC turns (i.e., pre-compression) the curve obtained in the IC0 condition shifts to the left and the BINS could present an axial stiffness ranging from 39 N/mm to 193 N/mm.

### 3.2. Sensor Calibration and Validation

The raw sensor output is shown in Figure 6 in which both the potentiometer raw output and the combined sensors are displayed for the flexion and lateral bending tests. The resulting calibration matrix was applied to the sensors and implemented in a static and dynamic validation test, which is shown in Figure 7. Compared to Motion Capture, BINS sensors managed to provide a measure of the angle with RMSE values of 1.35° and 1.31° for the flexion/extension and lateral bending sensors, respectively. The sensors maintained a good angle estimation in the impact test, where the RMSE was 2.99° and 1.29° for the flexion/extension and lateral bending sensors, respectively.

In the axial test, the voltage output and the axial compression were well-fitted by the linear model, with an R2 value of 0.998. The sensitivity of the sensors in compression was of 0.3045 mm/V (with an excitation voltage of 5 V).

## 4. Discussion

This study aimed to develop a modular neck that is useful for analyzing crash and impact tests and for providing a comparative assessment of neck protective gear performance. This implies that the neck should be biofidelic in terms of its mechanical behavior and that it needs to provide feedback from sensors.

Regarding the first objective, the neck design targeted data from cadaver vertebrae to build the FSU. This was accomplished by properly choosing the properties of the surrogate disc positioned between the vertebral bodies. However, it is debatable whether a human neck, even of an unconscious subject, would respond as the plain vertebrae in [8], in which any other structure except the bone, the disc, and intervertebral ligaments were rejected. Indeed, material properties were chosen to obtain a dummy neck that was slightly stiffer than the above cadaver studies. This allowed reaching the stiffness of either a complete cadaveric neck or a passive human subject [6,7], after the addition of the inner cable in the prototype.

Compared to other surrogates, the BINS has much more flexibility than the Hybrid III, which is two orders of magnitudes stiffer, and falls close to LUSN neck and passive human data [9,14]. To achieve proper asymmetry of the human neck, O-Rings were added to special hooks, and the stiffness was effectively modified on certain planes and load directions of the surrogate. In addition, modifying the stiffness properties of the O-Rings or the stiffening elements placed on the side hooks leads to different stiffening effects. Indeed, the results in Table 1 and Appendix A show that the same stiffness range can be achieved either by tensioning the IC or using an appropriate number of O-Rings. Moreover, using the second approach, the neck achieved the desired biofidelic asymmetric behavior. Therefore, a combination of these two factors can be used to adjust the neck bending behavior according to the test requirements. Based on the experiments and the comparison with literature data, it appears that the IC0+ORings condition reported in Table 1 could be used to simulate a passive human flexural response [6].

Axial response of the neck can also be a concern for certain testing protocols, and the characterization of the BINS axial stiffness was addressed. The current prototype is also in this case far more compliant than the Hybrid III, which is itself too stiff when compared to cadaveric human data [21]. To be fair, the BINS falls slightly under the human cervical column curve of Figure 5, and has a similar behavior only when a pre-compression (IC6 condition) is applied.

Finally, the torsional response of the neck was also evaluated. This aspect could be less critical in most of applications, since protectors mainly focus on the other two anatomical directions. Anyway, results showed that the BINS outstands the Hybrid III in terms of biofidelity with cadaver neck. Nevertheless, the BINS is still quite far from the neck of a human cadaver, which responds almost freely up to about 50° before stiffening. Conversely, the response of the BINS is more linear.

In conclusion, the presented prototype well approximates a passive human both in bending and compression. Nevertheless, a change of the rubber with materials having higher shore hardness can tune the neck to target other human conditions. Moreover, having a modular design such as the BINS has several advantages over existing solutions like Hybrid III or other beam-like surrogates. Indeed, it is possible to tune each FSU material differently and obtain the proper heterogeneous stiffness of the biological neck; it is also possible to interpose wedges between the FSU obtaining a more physiological static curvature like in other surrogates [13]. This design is also faster and more cost-effective to repair, since damaged modules can be replaced independently. The hooks represent another important aspect of the design and not only provide a way to increase the stiffness with the O-Rings, but can also be used as insertion points for muscles and tendons surrogates that could be included in future versions of the BINS.

For the retrieval of useful angular and kinetic data, the presented dummy neck has four embedded rotary potentiometers that, thanks to a pulley system, can estimate the angle on the two main flexion axes. Specifically, during a combined rotation test, an RMSE of 1.35° was found over a 70° range of motion. The same sensors can be effectively used to measure neck compression by combining their outputs. Moreover, the neck features the possibility of adding a six-axis load cell to the occipital level and is compatible with Hybrid III load cells for both the upper and lower neck.

The described surrogate is an alternative for testing neck protective gear and can provide a full kinematic and kinetic evaluation without the support of external cameras or other devices. Moreover, preliminary tests were performed using a seated dummy and a commercial motorbike neck brace in rear pendulum impacts, obtaining an initial validation of the surrogate and its sensors, which still approximated the main angle with an error of 3° over 50° in the dynamic test.

Besides showing some interesting results, the prototype still has some limitations that could leave room for future improvements. The current version showed agreement with passive human data, but the material of stiffer additional elements on the hooks may need to be changed to simulate humans with higher degree of muscular activation. Another point that needs to be furtherly investigated is the durability of the neck under repeated tests and its resistance to high-energy impacts, since its crashworthiness was tested only in a low-energy protected impact. Therefore, the need and feasibility for stronger materials (e.g., aluminum vertebrae) and bonding to the rubber should be evaluated. Finally, the kinematic sensors still need some validation (e.g., under long-term drift, durability, bandwidth) and improvement, such as the inclusion of a torsional sensor.

Future work will aim to address the above limitations by testing the structural properties of the developed neck in more detail to guarantee its applicability to high-energy or full-scale testing or both with ATDs. Indeed, while the BINS proved strong enough to withstand protected and low-energy impacts, its crashworthiness in different situations is unknown and possibly still insufficient. A possible solution could be the substitution of the plastic elements with aluminum powder-sintered components. This change in material may also be paired with another technical solution for the bonding with the rubber to obtain a stronger version of the BINS. The biofidelity could be further improved by increasing the non-linearity with different stiffening solutions. A possible solution to provide stiffening could be to wrap the surrogate with a silicone skin anchored to each hook, which would increase stiffness evenly along the surrogate. The bench test results could be validated using other setups, such as a comparison of the global head kinematics [13]. Another challenge is the development of an axial torsion sensor that is lacking in the current prototype, which could benefit by recent approaches using smart clothing technology for wearable 3D angular sensors [22]. Together with these improvements, the next steps would also benefit from testing the repeatability between different copies of the surrogate, which is a key parameter towards its broader use.

## Figures and Tables

**Figure 1 sensors-25-04925-f001:**
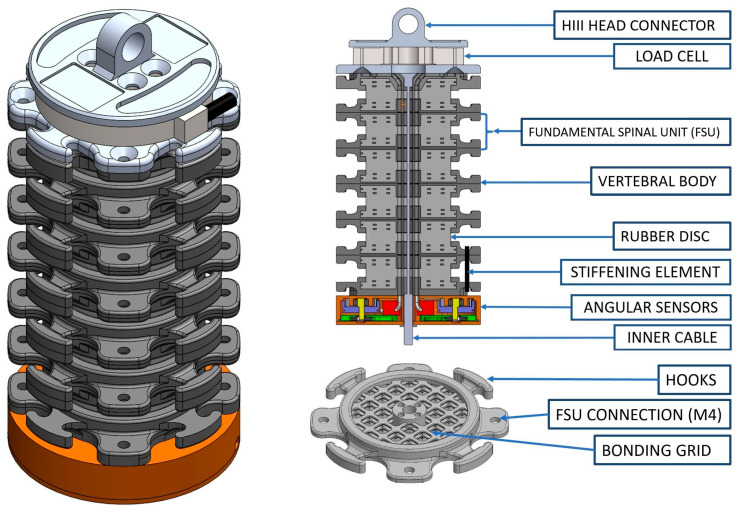
Assembly of the BINS. The main components of the BINS are described and indicated. It is also possible to see the bonding grid which is used to securely join the rubber and the plastics.

**Figure 2 sensors-25-04925-f002:**
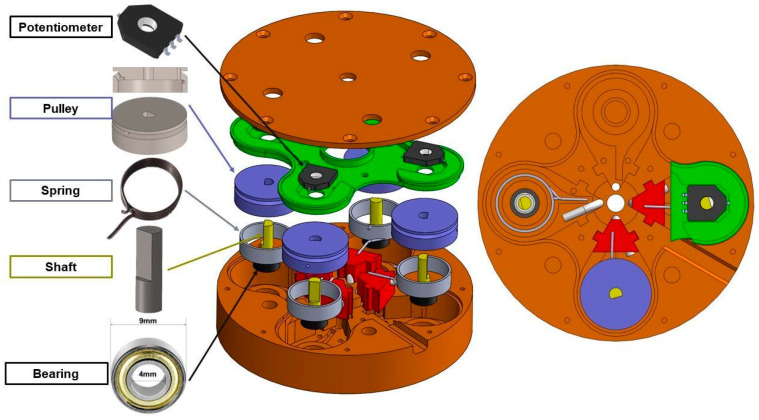
Description of the BINS sensors. The components are showed in an upside-down exploded view, and bottom view. Wires coming from the neck are re-directioned thanks to a curved conduct (red block) which bends the PTFE sheath (white) before entering in the spring-loaded pulley system.

**Figure 3 sensors-25-04925-f003:**
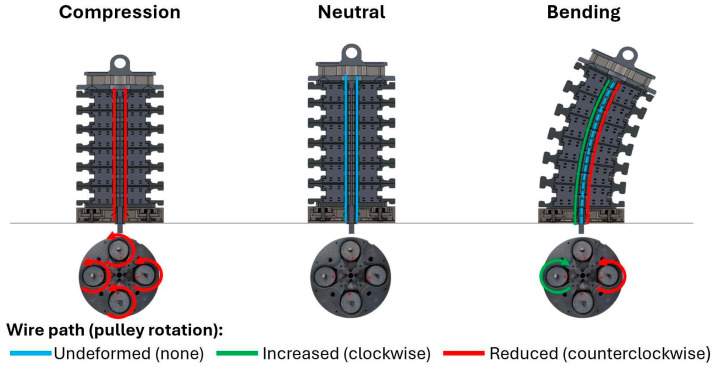
Modification in length of the cable conduits during basic movements of the BINS causing the inextensible wires to rotate the spring-loaded pulleys changing potentiometers output and allowing angular deflection or compression measurement.

**Figure 4 sensors-25-04925-f004:**
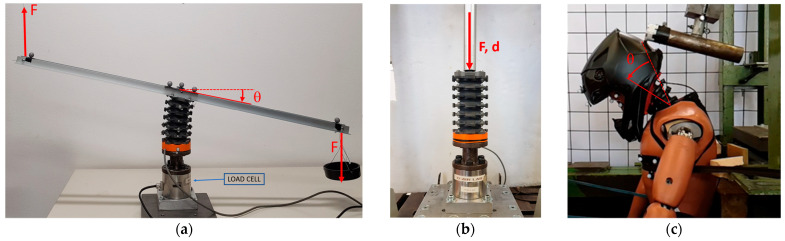
Testing apparatus for the characterization of the BINS. Different configurations are showed: (**a**) bending test setup inspired by [8] for bending, in the torsional test the setup is identical but the two forces are directed orthogonally to the view; (**b**) compression test setup, the cylindrical beam connects to the cylinder to compress the neck; (**c**) impact test, the dummy head was impacted with a steel pendulum on the back, reflective markers and integrated sensors capture the angle of the neck.

**Figure 5 sensors-25-04925-f005:**
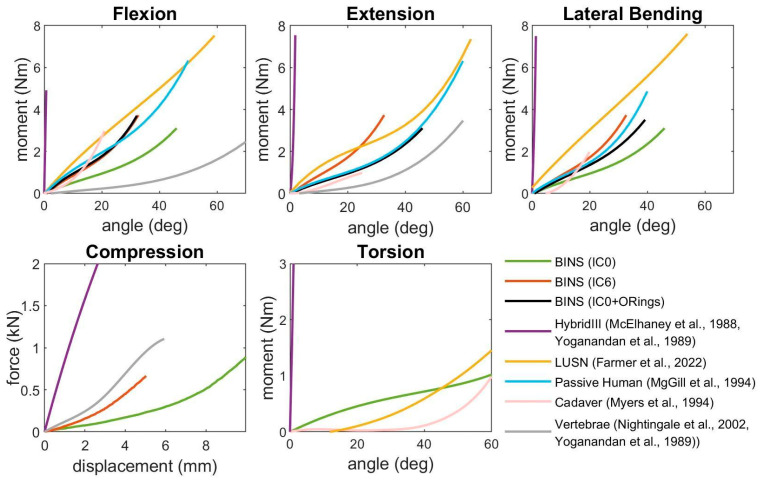
Results of the characterization test. Data of the BINS is compared to literature data in different planes and loading conditions. In compression test the response is not dependent on the OR disposition, and in torsional test is solely dependent on the rubber properties.

**Figure 6 sensors-25-04925-f006:**
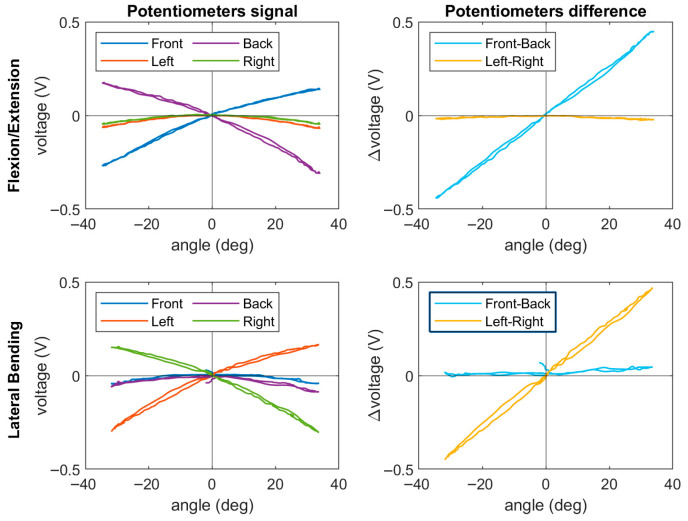
Potentiometer output during the calibration tests. The output of each single sensor on the left is combined by subtraction of the opposite potentiometers, obtaining an amplified and straighter signal.

**Figure 7 sensors-25-04925-f007:**
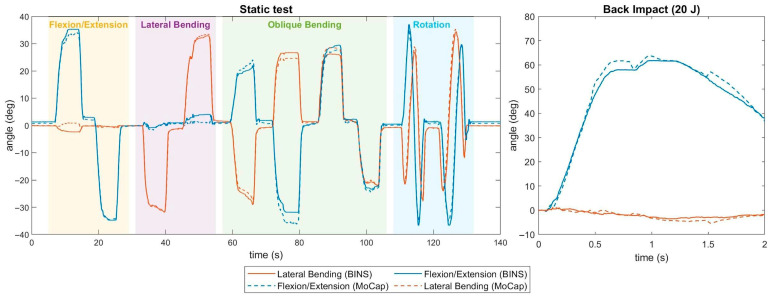
Proof of the functioning of the angular sensors. Data collected by the potentiometers are compared with motion capture in: static test (**left**), the neck is moved by hand in different ways, highlighted with shaded areas in the chart; dynamic test (**right**), the neck flexes in the sagittal plane due to a back impact.

**Table 1 sensors-25-04925-t001:** Surrogate neck stiffness computed during bending test, and comparison with literature data of other surrogates and human subjects.

Neck Surrogate/ Human Data	Flexion Stiffness [Nm/rad]	Extension Stiffness [Nm/rad]	Lateral Stiffness [Nm/rad]
IC0	3.29	3.29	3.29
IC6	5.64	5.64	5.64
IC0+O-rings ^1^	5.76	3.29	4.46
Hybrid III [9]	393.24	212.91	306.00
LUSN [14]	7.24	5.82	8.11
Passive Human [6]	6.48	4.72	5.95
Cadaver [7]	7.00	2.15	4.76
Vertebrae [8]	1.88	2.44	N/A

^1^ two O-rings for each posterior and lateral hook, which are activated during flexion and lateral bending.

## Data Availability

The raw data supporting the conclusions of this article will be made available by the authors on request.

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
