# Peer review of "Development of the Biofidelic Instrumented Neck Surrogate (BINS) with Tunable Stiffness and Embedded Kinematic Sensors for Application in Static Tests and Low-Energy Impacts"

_sensors, 2025, doi:10.3390/s25164925_

Round 1

Reviewer 1 Report

Comments and Suggestions for Authors

Title: Development of the Biofidelic Instrumented Neck Surrogate (BINS) with tunable stiffness and embedded kinematic sensors for application in static tests and low-energy impacts.

This paper describes a modular neck surrogate (BINS) with adjustable stiffness and built-in sensors, replicating human-like behavior for accurate, low-energy impact testing of protective gear. Results show that it replicates human-like neck behavior and provides embedded kinematic measurements that offer a valuable tool for evaluating protective gear. However, some details of the manuscript need to be refined. On this basis, a slight modification is required before the publication can be accepted.

  1. I recommend that you try not to use abbreviations in the legends of images, as they can make it difficult to read images, such as the FSU in Figure 1.
  2. While the modular design and tunable stiffness are strengths, I suggest that the paper should more clearly explain how the chosen rubber and 3D-printed materials were selected to match human cervical biomechanics, and discuss any trade-offs in fidelity or durability.
  3. The current validation focuses on low-energy impacts, but many real-world scenarios involve higher forces. I suggest that the authors should explicitly discuss how the design could be adapted or reinforced for such tests, or clarify its intended scope.
  4. The language skill of all this paper should be improved.
  5. The citations of the paper are not many and some good papers can be cited, such as eScience: 100373. Nano Energy, 113: 108574. Exploration 4(1): 20230109.

Author Response

Comments 1: This paper describes a modular neck surrogate (BINS) with adjustable stiffness and built-in sensors, replicating human-like behavior for accurate, low-energy impact testing of protective gear. Results show that it replicates human-like neck behavior and provides embedded kinematic measurements that offer a valuable tool for evaluating protective gear. However, some details of the manuscript need to be refined. On this basis, a slight modification is required before the publication can be accepted.

Response 1: We are glad to receive your positive comments, we did our best to revise the manuscript according to your suggestions. Please find the changes in the track_changes version of our manuscript.

Comments 2: I recommend that you try not to use abbreviations in the legends of images, as they can make it difficult to read images, such as the FSU in Figure 1.

Response 2: Thanks for the comment, we explicated Fundamental Spinal Unit in Figure 1

Comments 3: While the modular design and tunable stiffness are strengths, I suggest that the paper should more clearly explain how the chosen rubber and 3D-printed materials were selected to match human cervical biomechanics, and discuss any trade-offs in fidelity or durability.

Response 3: Thanks for the comment, we added some more clarification in the methods section (L160-L167).

Comments 4: The current validation focuses on low-energy impacts, but many real-world scenarios involve higher forces. I suggest that the authors should explicitly discuss how the design could be adapted or reinforced for such tests, or clarify its intended scope.

Response 4: Dear Reviewer, we already tried to take this into consideration in the final paragraphs of the discussion, we furtherly stressed on this point to make it clearer for the readers (L437-L452).

Comments 5: The language skill of all this paper should be improved.

Response 5: Thanks for the comment, we furtherly revised the English of the paper to improve it.

Comments 6: The citations of the paper are not many and some good papers can be cited, such as eScience: 100373. Nano Energy, 113: 108574. Exploration 4(1): 20230109.

Response 6: Thanks for the comment, we added some more references including one of these that you suggested (L458-460).

Reviewer 2 Report

Comments and Suggestions for Authors

In this study, the authors developed a Biofidelic Instrumented Neck Surrogate (BINS), a modular neck surrogate with tuneable stiffness and embedded kinematic sensors. When integrated onto the Hybrid III dummy, the BINS was able to mimic human neck flexural behaviour in low-energy impacts. Therefore, the study provided a valuable tool for impact analysis and protective gear evaluation. Overall, the manuscript is very good in novelty, methodology, and writing.

There are some minor issues that if addressed will further improve the quality of the research.

  1. The wires were made of coated braided fishing lines, and in this research they were considered inextensible. This assumption may be too strong. What was the tensile modulus of the wires? Could you provide the stress-strain curves?
  2. The material properties of the surrogate disc were targeted to obtain a neck bending response similar to that in cadaveric experiments. So, what were the materials properties? It is necessary to clearly list the parameters, as they are crucial to make the FSUs more biomimetic.
  3. In the experiments, reflective markers have been used to establish the reference system. Can you provide the pictures of the markers and their specific locations?
  4. It is suggested to evaluate the repeatability of the BINS in your future study, at least in static or quasi-static conditions.

Author Response

Comments 1: In this study, the authors developed a Biofidelic Instrumented Neck Surrogate (BINS), a modular neck surrogate with tuneable stiffness and embedded kinematic sensors. When integrated onto the Hybrid III dummy, the BINS was able to mimic human neck flexural behaviour in low-energy impacts. Therefore, the study provided a valuable tool for impact analysis and protective gear evaluation. Overall, the manuscript is very good in novelty, methodology, and writing.

There are some minor issues that if addressed will further improve the quality of the research.

Response 1: We are glad to receive your positive comments, we did our best to fix the issues you pointed out and make and improve the quality of the work. Please find the changes in the track_changes version of our manuscript.

Comments 2: The wires were made of coated braided fishing lines, and in this research they were considered inextensible. This assumption may be too strong. What was the tensile modulus of the wires? Could you provide the stress-strain curves?

Response 2: Thanks for the comment, we try to explain our assumption. Unfortunately, we don’t have this kind of data for the wires neither the possibility to setup a proper tensile test to investigate their elastic properties very soon. However, on a qualitative basis, the wire is far stiffer than the torsional spring which is linked to in a series of springs. Therefore, for a given movement of the neck (e.g., bending like in Fig 3), which requires the free length of the fishing line to increase/decrease we considered the elongation of the wire negligible compared to the rotation of the pulley needed to feed/return wire. Anyway, if this assumption was incorrect, the elongation of the wire will still be considered and compensated by the calibration coefficients. We described the basis for our hypothesis more in detail and also provided info its implications in the methods (L205-L212).

Comments 3: The material properties of the surrogate disc were targeted to obtain a neck bending response similar to that in cadaveric experiments. So, what were the materials properties? It is necessary to clearly list the parameters, as they are crucial to make the FSUs more biomimetic.

Response 3: Thanks for the comment, we were mainly interested in the elastic properties of the material. We did some testing in preliminary works by using rubbers of different hardness (usually provided by the manufacturer) and we decided to build the full prototype using Platsil Gel 10 rubber, which has a Shore value equal to A10. We described this in the manuscript (L163-L167).

Comments 4: In the experiments, reflective markers have been used to establish the reference system. Can you provide the pictures of the markers and their specific locations?

Response 4: Dear reviewer, the position of the markers is already showed in Figure 4a. However, thanks to your suggestion, we clarified and described their position in the manuscript (L237-241).

Comments 5: It is suggested to evaluate the repeatability of the BINS in your future study, at least in static or quasi-static conditions.

Response 5: Dear reviewer, thanks for this precious suggestion. This, together with other works on the surrogate is already in our plans. We added a final sentence making this clear also to the reader (L459-L461).

Reviewer 3 Report

Comments and Suggestions for Authors

The present work aimed to develop a biofidelic dummy neck having a bending, torsional and axial response similar to human cervical spine. The availability of a biofidelic neck substitute is of great importance because it can provide an accurate prognosis of spinal injuries and is also responsible for the correct kinematics of the head.

The paper is well written and covers the problem of neck traumas, which is relevant at all times. At the same time, the authors refer to works published more than 10 years ago. It is necessary to update the literature references on the problem of neck injuries and methods of studying kinematics so that the article is more modern and relevant.

Some improvements in Methodology and Results is needed.

The Methodology section must specify the number of iterations for validation of the device.

The results (table 1) must indicate not only the average measurement values, but also the interval.

The authors need to present the main result of the design of the prototype of the neck substitute more clearly. Which model from Table 1 more accurately reflects the studied kinematic properties of the neck?

It is necessary to provide evidence of the advantage of the prototype being created compared to existing models. It is necessary to make a statistical analysis of the obtained results.

The abstract needs to be structured. It is necessary to clearly indicate the purpose of the study, the methods, devices and equipment used. The main result obtained by the authors should be presented.

Overall conclusion: the manuscript is not ready for publication. Revision of the maintext and improvement of the methodology is necessary to increase the readability and relevance of the study.

Author Response

Comments 1: The present work aimed to develop a biofidelic dummy neck having a bending, torsional and axial response similar to human cervical spine. The availability of a biofidelic neck substitute is of great importance because it can provide an accurate prognosis of spinal injuries and is also responsible for the correct kinematics of the head.

Response 1: We are glad that the reviewer agrees with us on the need of having accurate neck surrogates for testing, we did our best to improve our paper by addressing the reviewer’s comments.

Comments 2: The paper is well written and covers the problem of neck traumas, which is relevant at all times. At the same time, the authors refer to works published more than 10 years ago. It is necessary to update the literature references on the problem of neck injuries and methods of studying kinematics so that the article is more modern and relevant.

Response 2: Dear reviewer, thanks for this precious suggestion. We added two more references to put more in context the problem. We hope that this fulfills your request, but we are open to further suggestions.

Comments 3: Some improvements in Methodology and Results is needed.

Response 3: Thanks for the comment, we did some improvements to the methodology also thanks to the other reviewers’ comments.

Comments 4: The Methodology section must specify the number of iterations for validation of the device.

Response 4: Dear reviewer, we added a sentence in the methods (L161-L167) about intermediate steps for choosing the rubber to get a proper response. Regarding the full prototype object of the paper, this was only built one based on previous experience.

Comments 5: The results (table 1) must indicate not only the average measurement values, but also the interval.

Response 5: Dear reviewer, we designed our test to investigate the effect of inner cable (IC) and orings (OR) on the final stiffness. Since we have just one prototype, we chose to perform each test once. Otherwise, we could have mistakenly exchanged sample numerosity and repeatability of the same sample in the ANOVA calculations. We did not investigate repeatability of the single test, but since we considered multiple load steps and data points for the stiffness calculation we are confident that errors in readings and other factors could be averaged by this. Table 1 reports the stiffness value obtained by each of the tested conditions, and for the above discussed motivations, it is a single value rather and average and SD, which would require multiple prototypes to be built and tested.

Comments 6: The authors need to present the main result of the design of the prototype of the neck substitute more clearly. Which model from Table 1 more accurately reflects the studied kinematic properties of the neck?

Response 6: Dear reviewer, there is not a single answer to this question. We could say that the IC0+ORings configuration (the black line in figure 5) best fits the Passive human data that we found in literature. Instead, if the interest is in compression, the IC6 condition is closer to the stiffness of the vertebrae. We indicated these possible choices in the discussion (L395-L397 and L401-403).

Comments 7: It is necessary to provide evidence of the advantage of the prototype being created compared to existing models. It is necessary to make a statistical analysis of the obtained results.

Response 7: Dear reviewer, we are aware that other surrogates exist and can be useful tools with good biofidelity as clearly visible in Figure 5. We think that our advantages are the tunability of the stiffness and also the presence of embedded angular sensors, which both allow ours surrogate to span multiple moment-angle curves in Figure 5 and provide feedback in static and dynamic tests. In the results section we reported a statistical analysis of the change of stiffness with respect to change in stiffening elements for our surrogate, but we do not have the necessary data to make a statistical analysis comparing our surrogate with the others in literature.

Comments 8: The abstract needs to be structured. It is necessary to clearly indicate the purpose of the study, the methods, devices and equipment used. The main result obtained by the authors should be presented.

Response 8: Dear reviewer, thanks for this precious suggestion. We rewritten the abstract and modified its structure according to your request.

Comments 9: Overall conclusion: the manuscript is not ready for publication. Revision of the maintext and improvement of the methodology is necessary to increase the readability and relevance of the study.

Response 9: Dear reviewer, we understand your evaluation. We hope that the improvements done to the work thanks to the precious comments from you and the other reviewers made you change your initial conclusion.

Round 2

Reviewer 3 Report

Comments and Suggestions for Authors

the manuscript has been sufficiently improved to warrant publication in Sensors